# *Trifolium Repens* Blocks Proliferation in Chronic Myelogenous Leukemia via the BCR-ABL/STAT5 Pathway

**DOI:** 10.3390/cells9020379

**Published:** 2020-02-06

**Authors:** Federica Sarno, Giacomo Pepe, Pasquale Termolino, Vincenzo Carafa, Crescenzo Massaro, Fabrizio Merciai, Pietro Campiglia, Angela Nebbioso, Lucia Altucci

**Affiliations:** 1Department of Precision Medicine, University of Campania “Luigi Vanvitelli”, 80138 Naples, Italy; federica.sarno@unicampania.it (F.S.); vincenzo.carafa@unicampania.it (V.C.); crescenzo.massaro@unicampania.it (C.M.); 2Department of Pharmacy, School of Pharmacy, University of Salerno, 84084 Fisciano, Italy; gipepe@unisa.it (G.P.); fmerciai@unisa.it (F.M.); pcampiglia@unisa.it (P.C.); 3Institute of Biosciences and Bioresources (IBBR), National Research Council of Italy (CNR), 80055 Portici, Italy; pasquale.termolino@ibbr.cnr.it; 4PhD Program in Drug Discovery and Development, University of Salerno, 84084 Fisciano, Italy; 5European Biomedical Research Institute of Salerno, 84125 Salerno, Italy

**Keywords:** *Trifolium repens*, isoflavonoids, polyphenols, cancer, chronic myelogenous leukemia, K562 cells, BCR-ABL/STAT5

## Abstract

Some species of clover are reported to have beneficial effects in human diseases. However, little is known about the activity of the forage plant *Trifolium repens*, or white clover, which has been recently found to exert a hepatoprotective action. Scientific interest is increasingly focused on identifying new drugs, especially natural products and their derivatives, to treat human diseases including cancer. We analyzed the anticancer effects of *T. repens* in several cancer cell lines. The phytochemical components of *T. repens* were first extracted in a methanol solution and then separated into four fractions by ultra-high-performance liquid chromatography. The effects of the total extract and each fraction on cancer cell proliferation were analyzed by MTT assay and Western blotting. *T. repens* and, more robustly, its isoflavonoid-rich fraction showed high cytotoxic effects in chronic myelogenous leukemia (CML) K562 cells, with IC_50_ values of 1.67 and 0.092 mg/mL, respectively. The block of cell growth was associated with a total inhibition of BCR-ABL/STAT5 and activation of the p38 signaling pathways. In contrast, these strongly cytotoxic effects did not occur in normal cells. Our findings suggest that the development of novel compounds derived from phytochemical molecules contained in *Trifolium* might lead to the identification of new therapeutic agents active against CML.

## 1. Introduction

In 2018, the World Health Organization reported 9.6 million deaths due to cancer worldwide, indicating that the number of cancer patients is increasing greatly [1]. Leukemia is one of the most common tumors, with 8990 new cases and 1140 deaths from chronic myelogenous leukemia (CML) estimated to occur in the United States in 2019 (https://seer.cancer.gov/statfacts/html/cmyl.html) [2]. CML is a hematopoietic disease that mainly affects men (58% men and 42% women) between 50 and 60 years of age (https://www.cancer.org/cancer/chronic-myeloid-leukemia/about/statistics.html) [3], and is characterized by an abnormal proliferation and accumulation of mature granulocytes. In 90–95% of patients, CML is caused by a genetic alteration known as Philadelphia (Ph) chromosome, which was discovered for the first time in 1973 [4]. This mutation involves the reciprocal gene translocation of chromosome t(9;22)(q34;q11), leading to a hybrid protein in which the amino-terminal sequence of the *BCR* gene is fused with the carboxy terminal of *ABL*. This translocation leads to the expression of BCR/Abl oncoprotein in two alternative splicing sites at 210 and 190 Da [5,6]. The BCR/Abl chimeric protein is constitutively active, showing a high tyrosine kinase activity that induces the expression of downstream pathways involved in cell survival and apoptosis inhibition, such as JAK/STAT, PI3K/AKT, and CRKL, and the reduction of pro-apoptotic factors, including p38 [7,8,9]. This chromosomic translocation is considered the primary event involved in CML progression. Inhibition of BCR/Abl expression is, therefore, the preferential strategy for CML treatment. Imatinib mesylate (IM) was the first drug to be developed for anti-CML therapy as it is able to directly target BCR/Abl [10]. However, approximately 33% of patients develop pharmacological resistance to IM (15–25% of cases caused by cytogenetic and 2–4% by hematologic resistance) [11,12]. Desatinib and nilotinib are two new-generation drugs used in CML therapy developed for imatinib-resistant or imatinib-intolerant disease treatment [13,14,15]. However, recent studies also describe desatinib and nilotinib resistance primarily caused by a genetic mutation [16,17,18]. There is, therefore, an urgent need to discover and characterize novel compounds that are useful in CML therapy.

High-throughput screening (HTS) is a common approach in drug discovery that is used by the pharmaceutical industry, as well as in academic research. HTS is based on the screening of a very large number of natural and synthetic modulators to rapidly characterize their toxicological effect(s) and biological target(s).

Recently, interest in discovering novel natural drugs for cancer treatment has significantly increased. In the last 20 years, 80% of anticancer substances approved by the Food and Drug Administration were natural products [19,20].

Tumor cells easily develop resistance to single drug treatments due to multifactorial regulation and not due to single molecular target alteration. Efforts by the scientific community to develop combination therapies for cancer treatment have, therefore, increased in recent years. By acting on two different targets simultaneously, combo treatments can produce a synergistic effect and reduce resistance to single drugs. Plants produce a variety of evolutionary conserved molecules differing in structure and activity, that are able to simultaneously act on multiple targets and produce a highly self-protective effect [21]. By exploiting these properties, pharmaceutical companies have over the years focused their drug discovery programs on characterizing natural drugs to be used alone or in combination in the treatment of several human diseases and syndromes [22,23,24,25].

Polyphenols are an important group of phytochemical compounds extensively used in medical treatments [26]. Consumption of foods rich in polyphenols were shown to prevent or delay the onset of several diseases by exerting an antioxidant effect, inhibiting ROS formation, or modulating the expression of genes and proteins involved in cell proliferation [27,28]. Based on their chemical structure, polyphenols can be divided into different classes, the most common of which are (**I**) flavonoids/isoflavonoids, (**II**) phenolic acids, (**III**) lignans, and (**IV**) stilbenes. Flavonoids are the most abundant class and high amounts are found in various species of clover [29,30,31].

Clover is a plant often used for forage and in medicine due to its beneficial activity [32,33]. *Trifolium alexandrine* and *Triflolium ruspinantus* were both shown to be effective against hepatotoxicity [34]. Specifically, *T. ruspinantus* protects cells from the hepatotoxic effect of phosphamide by attenuating oxidative stress and inflammation via increased levels of NRF2 [35]. The red clover *Trifolium pratense* displayed a robust therapeutic activity by reducing the proliferation of breast cancer cells [36].

*Trifolium repens* (TR), commonly known as white clover, is a perennial herbaceous plant native mainly to Europe and Central Asia. It is used as a common fodder crop for cattle and in some environments is used to absorb heavy metals from soil. TR might also have a potential role in atmospheric nitrogen fixation, as it contains nitrogen-fixing bacteria in its roots, which form nodules [37,38,39]. Nutritionally, it is a source of proteins and sugars, has a low fiber and high mineral content, and is rich in polyphenols. Unlike other clover species, little is known about the beneficial activity of TR. In some regions of Turkey, TR is used as an expectorant, antiseptic, and analgesic. The hepatoprotective function of the aqueous phenolic fraction extracted from TR was only very recently demonstrated [40].

In this study, we assessed the antitumor activity of TR on a panel of liquid and solid cancer cell lines, including colon cancer HCT-116, breast cancer MCF7, lung cancer A549, and hepatocellular carcinoma HepG2 cells, and observed an effect only in CML cells. After separation and isolation of the isoflavonoid fraction, we found that this molecule group affected all tested leukemia cell lines, but with greater specificity on CML cells, inhibiting the BCR/Abl expression and oncogenic proteins involved in cancer progression.

## 2. Materials and Methods

### 2.1. Trifolium Repens Component Extraction

#### 2.1.1. Plant Material

A quantity of 0.1 g of white clover seeds (obtained from CNR-IBBR, UOS Portici) was sown in individual pots (24 cm long, 15 cm wide, and 8 cm deep) filled with sterilized quartz sand. The seeds germinated in growth chambers (day/night temperatures of 21/18 °C and 790 μmol m^−2^ s^−1^ photosynthetically active radiation), for seven days. Plantlets were irrigated with water for another 25 days under the same growth conditions. Adult plants were harvested, freeze dried, and pulverized with mortar and pestle.

#### 2.1.2. Polyphenol Extraction

Polyphenols were extracted following a previously published procedure [37] with some modifications. Briefly, 25 mg of pulverized samples were extracted in 1.5 mL of 75% (*v*/*v*) methanol containing 0.05% (*v*/*v*) trifluoroacetic acid (TFA), and then incubated at room temperature for one hour, in a continuous orbital shaker at a medium speed. After incubation, the samples were centrifuged at 19,000 × *g* for 10 min. The extracts were filtered through 0.2 mm polytetrafluoroethylene filters. The filtered extract was concentrated in a Vacufuge Concentrator (Eppendorf, Hamburg, Germany) and lyophilized. The powder was then resuspended in DMSO:H_2_O (9:1) at a final concentration of 100 mg/mL. The extraction yield was calculated as the weight ratio of the final lyophilized powder to the dried raw plant material used for the extraction.

### 2.2. RP–UHPLC–MS/MS and LC–MS/MS

#### 2.2.1. Instruments

RP–UHPLC–MS/MS analyses were carried out using a Shimadzu Nexera system, consisting of a CBM-20 controller, four LC-30AD reciprocating high-pressure piston pumps, a DGU-20 Ar5 degasser, a SIL-30 AC autosampler, a CTO-20AC column oven, and a photo diode array SPD-M20A (Shimadzu, Kyoto, Japan). The UHPLC system was coupled online with an Ion Trap-Time of Flight (IT-TOF) hybrid mass spectrometer, equipped with an electrospray source (ESI; Shimadzu). LC–MS/MS data were processed using the LCMSsolution^®^ software (Version 3.50.346, Shimadzu).

#### 2.2.2. RP–UHPLC–PDA–ESI–IT–TOF

In detail, the analyses were conducted using a Kinetex^®^ EVO C18 150 × 2.1 mm (100 Å) column, with a 2.6 μm core shell particulate (Phenomenex, Bologna, Italy). The flow of the mobile phases was set at 0.5 mL/min and the oven temperature was set at 45 °C. The injection volume was 5 µL. The analyses were carried out using H_2_O (A) and acetonitrile (ACN) (B), both acidified at 0.1% (*v*/*v*) acetic acid, using the following elution gradient: 0.01–3.00 min, isocratic at 2% B; 3.01–15.00 min, 2–25% B; 15.01–20.00 min, 25–60% B; 20.01–22.00 min, isocratic at 95% B and finally 4 min for column re-equilibration. The following photo diode array (PDA) detector parameters were applied—sampling rate, 12.5 Hz; detector time constant, 0.240 s; cell temperature, 40 °C. Data acquisition was set in the range of 190–800 nm and chromatograms were monitored at 280 nm and 254 nm, at the maximum absorbance of the compounds of interest.

MS experiments were performed as follows—flow rate from LC was split 50:50 prior to the ESI source by means of a stainless steel Tee union (1/16 in, 0.15 mm bore; Valco, Houston, TX, USA). Resolution, sensitivity, and mass number calibration of the ion trap and the TOF analyzer were tuned using a standard sample solution of sodium trifluoroacetate. After the calibrant had flowed, cleaning operation of the tube and ESI probe was carried out using flowing ACN (0.2 mL/min, 20 min).

MS detection was operated both in the positive and negative ionization mode, with the following parameters—detector voltage, 1.65 kV; curve desolvation line (CDL) temperature, 250 °C; block heater temperature, 250 °C; nebulizing gas flow (N2), 1.5 L/min; drying gas pressure, 95 kPa. Full scan MS data were acquired in the range of 150–1500 m/z (ion accumulation time, 25 ms; IT, repeat = 3). MS/MS experiments were conducted in a data-dependent acquisition, precursor ions were acquired in the range of 150–1000 m/z; peak width, 3 Da; ion accumulation time, 50 ms; collision induced dissociation (CID) energy, 50%; collision gas, 50%; repeat = 1; and execution trigger (BPC) intensity, at the 95% stop level.

### 2.3. Semiprep-RP–HPLC–UV/Vis

The extract was fractionated by semi-preparative reversed phase liquid chromatography. For separation, a Shimadzu Semiprep-HPLC was used, consisting of two LC 20 AP pumps, a SIL 20 AP autosampler, a fraction collector FRC 10A, a UV detector SPD 20 A, equipped with a preparative cell and a system controller CBM 20 A.

The separation was carried out on a Luna C18 250 × 10 mm × 5 μm (100 Å) column (Phenomenex, Bologna, Italy), flow rate 5 mL/min, injection volume 1 mL, detection UV 254 and 280 nm, and the collection was based on UV triggering signal. The optimal mobile phase consisted of H_2_O (A) and ACN (B), both acidified by acetic acid 0.1% (*v*/*v*). Analysis was performed in gradient elution as follows—0.01–3.00 min, isocratic at 2% B; 3.01–30.00 min, 2–85% B; 30.01–34.00 min, isocratic at 99% B; and finally 3 min for column re-equilibration. Four fractions were collected on the basis of their elution times and, thus, hydrophobicity.

### 2.4. Cell Lines

Cell lines were purchased from ATCC. HCT-116 (colon cancer), MCF7 (breast cancer), and A549 (lung cancer) cells were propagated in Dulbecco’s Modified Eagle’s Medium (Euroclone, Milan, Italy), with 10% fetal bovine serum (FBS; Euroclone), 2 mM L-glutamine (Euroclone), and antibiotics (100 U/mL penicillin, 100 mg/mL streptomycin; Euroclone). HepG2 (liver hepatocellular carcinoma), K562 (CML), U937 (histiocytic lymphoma), NB4 (acute promyelocytic leukemia), and Mepr2B (mesenchymal progenitor) cells were propagated in RPMI-1640 Medium (Euroclone) containing 4.5 g/L glucose (Euroclone) supplemented with 10% FBS (Euroclone), 100 U/mL penicillin–streptomycin (Euroclone), and 2 mM L-glutamine (Euroclone).

### 2.5. Reagents

Cells were treated with lyophilized TR samples (total and fractions) and SAHA (N’-hydroxy-N-phenyloctanediamide; Vorinostat; Merck, Darmstadt, Germany) dissolved in DMSO (Sigma-Aldrich, Milan, Italy). DMSO was used as control in all experiments.

### 2.6. Cell Death Analysis

The percentage of cell death was calculated by PI analysis. K562 cells were treated for 24 and 48 h with Fraction D at 0.5, 0.25, and 0.125 mg/mL, or with SAHA at 5 μM for 24 h (a positive control for the induction of cell death in our experimental system). After treatment, cells were collected, then centrifuged (1200 rpm for 5 min) and suspended in a solution containing 1X PBS (Euroclone) and 0.2 mg/mL PI (Sigma-Aldrich). The percentage of cell death was calculated using a BD Accuri™ Flow Cytometer Instrument (BD Biosciences, Milan, Italy).

Formononetin, Daphnoretin, and Medicarpin (7%, 14%, and 26% of Fraction D, respectively) were tested at 0.034 mg/mL, 0.07 mg/mL, and 0.13 mg/mL (concentrations corresponding to 7%, 14%, and 26% of 0.5 mg/mL of Fraction D) for 24 and 48 h. PI analysis was then performed as previously described.

### 2.7. Annexin V/PI Assay

Annexin V/PI staining assay was performed according to the supplier’s instructions (Dojindo Molecular Technology, distributed by Microtech, Italy). Briefly, cells treated with the extract at indicated concentrations for 24 h were centrifuged and suspended in Annexin V binding solution at the final concentration of 1 × 10^6^ cells/mL. A total of 100 μL of this suspension was transferred into a new tube, to which 5 μL Annexin V (FITC conjugated) and 5μL propidium iodide (PI) were added. The reaction was carried out for 15 min at room temperature. The results were acquired using FACS Calibur (BD Biosciences). Graphs show the experimental results of biological triplicates.

### 2.8. MTT Assay

Cell viability was determined using the standard MTT assay. A total of 2 × 10^4^ cells/well were plated in a 96-well plate and treated, in triplicates, with TR total extract and with Fraction A, B, C, and D at different concentrations for 24 and 48 h, depending on the experiment. Thiazolyl Blue Tetrazolium Bromide [3-(4,5-dimethylthiazol-2-yl)-2,5-diphenyltetrazolium bromide] (MTT; Sigma-Aldrich) solution was added at 0.5 mg/ml. After 3 h, for the adhesion cells the supernatant was simply removed, whereas for the suspension cells, the plate was first centrifuged. The purple formazan crystals were dissolved in iso-propanol (Carlo Erba Reagents, Cornaredo, Italy) and the absorbance was read at a wavelength of 570 nm with a TECAN M-200 reader (Tecan, Männedorf, Switzerland). IC_50_ values were calculated using the GraphPad Prism7 software.

### 2.9. Total Protein Extraction

K562 cells were treated with 0.5 mg/mL of total TR extract for 24 and 48 h. After treatment, the cells were harvested and washed twice with PBS (Euroclone). Cells were lysed in a phosphorylated protein extraction buffer containing H_2_O with 50 mM Tris HCl pH 7.4 solution, 1 mM EDTA, 1% Triton X-100 (*v*/*v*), 150 mM NaCl, 5 mM MgCl_2_, 1 mM EGTA, 2 mM phenylmethylsulfonyl fluoride, 1 mM sodium orthovanadate, and 1X protein inhibitor proteinase cocktail before use. The cells were then incubated with the extraction buffer for 15 min at 4 °C, centrifuged at 16,000 × *g* at 4 °C for 30 min, and the supernatant was recovered. The total phosphorylated protein was determined using a Bradford assay (Bio-Rad, Milan, Italy).

### 2.10. Western Blotting

Western blotting analysis was performed by loading 40 μg of extracts at different concentrations of polyacrylamide gels, depending on the antibody band. The antibodies used were: pY177-BCR (#39019; Cell Signaling Technology, Pero, Italy), PathScan^®^ Bcr/Abl Activity Assay (Phospho-c-Abl, Phospho-Stat5 and Phospho-CrkL Multiplex Western Detection Cocktail) (#5300; Cell Signaling Technology), pT180/Y182-p38 (#92115; Cell Signaling Technology), pY308-AKT (#9275S; Cell Signaling Technology), AKT (#9272; Cell Signaling Technology), pY694-STAT5 (ab32364; Abcam, Milan, Italy), STAT5 (ab209544; Abcam), Tubulin (sc-5286; Santa Cruz Biotechnology, Segrate, Italy), GAPDH (sc-47724; Santa Cruz Biotechnology), and Actin (sc-47778; Santa Cruz Biotechnology). Semi-quantitative analysis was performed using the ImageJ software.

### 2.11. RNA Extraction and RT–PCR

Total RNA was extracted with Trizol (Invitrogen, Monza, Italy) and 1 μg was converted into cDNA using SuperScript^®^ VILO™ cDNA Synthesis Kit (Invitrogen), according to the manufacturer’s instructions. RT–PCR was performed with 50 ng of cDNA template in a 25-μL total reaction volume (12.5 μL Bio-Rad iTaq Universal SYBR Green supermix (2X), 0.5 mM of each gene-specific primer, H_2_O up to volume). Reactions were carried out on a Bio-Rad CFX-96 real-time PCR system (Bio-Rad Laboratories, Segrate, Italy). Each reaction was run in triplicates. For amplification, the following primers were used: STAT5, forward (5′-CTGAACAACTGCTGCGTCAT-3′), and reverse (5′-GTGGACGATGACAACCACAG-3′); GAPDH, forward (5′-GGAGTCAACGGATTTGGTCGT-3′) and reverse (5′-GCTTCCCGTTCTCAGCCTTGA-3′); BCR-ABL, forward (5′-CCACTGGATTTAAGCAGAGTTCAA-3′), and reverse (5′-TCCAACGAGCGGCTTCAC-3′).

### 2.12. Image Analysis

Cell images were acquired after 24 and 48 h of treatment with Fraction D at the reported concentrations, using Cytation 5 Cell Imaging Multi-Mode Reader (BioTek Instruments, Colmar, France).

### 2.13. Statistical Analysis

Graphs showed in the figures, represent the mean of the three independent experiments, with an error bar indicating the standard deviation. Differences between the treated cells versus control cells were analyzed using the GraphPad Prism 8.0 software (GraphPad Software, Inc., San Diego, CA, USA). Statistical comparison was performed by applying one-way analysis of variance (ANOVA) and Dunnett’s multiple-comparison test. Differences between groups were considered to be significant at a *p* value of <0.05.

## 3. Results

### 3.1. Trifolium Repens Inhibits Cancer Proliferation

TR (Figure 1A) was first grown and the total polyphenol fraction was then extracted by adapting a previously published protocol [41]. Briefly, pulverized samples of TR were extracted in 75% (*v*/*v*) methanol in water solution, containing 0.05% (*v*/*v*) trifluoroacetic acid (TFA) and, after concentration and lyophilization, were resuspended in DMSO:H_2_O (9:1) solution. To evaluate its anticancer activity, the TR total extract was tested at five different concentrations ranging from 1.0 mg/mL to 0.065 mg/mL for 48 h, in both solid and hematological cancer cells (Figure 1B–H and Appendix A). Treatment with TR did not affect the viability of solid cancer cell lines tested. Specifically, even at the highest concentration, TR did not alter the viability of colorectal cancer HCT-116 cells (Figure 1B), breast cancer MCF7 cells (Figure 1C), and hepatocellular carcinoma HepG2 cells (Figure 1D), while only a slight reduction in viability (~20%) was detectable in non-small lung cell cancer A549 cells (Figure 1E), compared to the DMSO-treated sample (used as control). However, interesting results were obtained in the CML K562 cells (Appendix A), in which the TR total extract reduced cell proliferation by about 50% and 80%, at 0.5 mg/mL and 1 mg/mL concentrations, respectively. These initial findings prompted us to hypothesize a potential selective anticancer action for TR in hematological malignancies.

To better investigate its anticancer selectivity, the total polyphenol extract of TR was tested in histiocytic lymphoma U937 (Figure 1F) and acute promyelocytic leukemia NB4 (Figure 1G) cells. After treatment for 48 h at different concentrations from 2 mg/mL to 0.065 mg/mL, cell viability was evaluated by MTT assay. Although NB4 cells are normally more sensitive to chemotherapeutic treatments, no significant effect was observed in either of these two cancer cell lines. Importantly, TR showed good cytotoxicity in K562 cells, with an IC_50_ value of 1.67 mg/mL at 48 h of treatment (Figure 1H), confirming our previous data.

Taken together, these results showed that the antileukemic activity of TR is limited to CML cells.

### 3.2. Trifolium Repens Inhibits BCR-ABL/STAT5 Signaling Pathway in Chronic Myelogenous Leukemia Cells

Since cell survival signaling pathways in the CML cells are modulated by BCR/Abl hybrid protein [42], we first investigated the effect of TR on BCR/Abl expression. Tyrosine 177 of BCR protein plays a critical role in activation of the anti-apoptotic pathway, such as AKT, and in transformation of hematopoietic progenitor cells in CML [43]. K562 cells were treated with TR at 0.5, 0.25, and 0.125 mg/mL concentration for 24 h, and BCR-ABL tyrosine 177 (anti-pY177AB) was evaluated. Only TR at 0.5 mg/mL was able to strongly reduce phosphorylation of BCR, and reduce the level of other proteins involved in cell proliferation, such as anti-pY207CRKL, anti-pY308AKT, and anti-AKT (Appendix A). K562 cells were then treated with 0.5 mg/mL TR total extract for 24 and 48 h. Treatment with TR also strongly reduced constitutive phosphorylation of BCR in both tyrosine 247 (anti-pY247AB) and 177 at 48 h (Figure 2A). In addition, the other crucial constitutive player involved in K562 cell survival, STAT5 phosphorylated in tyrosine 694 (anti-pY694STAT5), resulted in significant inhibition at 24 h of treatment. Interestingly, BCR and STAT5 inhibition was associated with lower levels of anti-pY207CRKL and with the total absence of anti-pY308AKT, after 24 h of exposure (Figure 2A,B).

After 48 h of TR treatment, phosphorylation levels of anti-pY694STAT5 and anti-pY308AKT increased slightly, unlike the total protein levels, which decreased at both induction times. The possible degradation of some components of the extract might have again led to an increase in phosphorylation residues.

Additionally, an 11- and 65-fold time-dependent increase (at 24 and 48 h, respectively) of anti-pThr180/Tyr182p38 compared to the control, was observed after TR exposure.

These results indicate that the anticancer activity of TR total extract is mediated by BCR-ABL/STAT5 pathway inhibition and p38 pathway activation.

### 3.3. Fraction D of Trifolium Repens Is Responsible for Its Antiproliferative Effects

To identify the fraction responsible for its anticancer activity, TR total extract was fractioned using an optimized chromatographic gradient. The total polyphenol extract composition was first analyzed using a Kinetex^®^ EVO C18 150 × 2.1 mm (100 Å) column, in H_2_O and acetonitrile (ACN), both acidified at 0.1% (*v*/*v*) of acetic acid solution, following the elution gradient—0.01–3.00 min, isocratic at 2% B; 3.01–15.00 min, 2–25% B; 15.01–304 20.00 min, 25–60% B; 20.01–22.00 min, and isocratic at 95% B (Appendix A). The total extract was then fractionated into four different fractions using semiprep-reversed phase chromatography on high-performance liquid chromatography (RP–HPLC), based on their hydrophobicity and, thus, elution times (about 5 min for each fraction). These four fractions were screened for their cytotoxicity by MTT assay in K562 cells treated at different doses, ranging from 2.0 mg/mL to 0.031 mg/mL for 24 and 48 h (Figure 3A–D). No toxic effect was observed after treatment with Fractions A and B, while Fraction C induced a significant proliferative block, only after 48 h of exposure to the two highest doses (2.0 and 1.0 mg/mL). Only Fraction D was found to exert a significant activity. Specifically, Fraction D induced a strong proliferative arrest in K562 cells at 0.25 mg/mL, at both 24 and 48 h. At 48 h of treatment, Fraction D showed an anticancer activity 10-fold greater than the total extract, with an IC_50_ value of 0.092 mg/mL (Figure 3D).

K562 propidium iodide (PI) analysis, in which the well-known chemotherapeutic SAHA was used as a positive control at 5 µM for 24 h, confirmed that at 0.25 mg/mL, Fraction D induced significant cell death (Figure 3E and Appendix A). Approximately 90% of PI-positive K562 cells were already detectable after 24 h of treatment (Figure 3E). To better characterize cell death induced by Fraction D, Annexin V/PI assay was performed after 24 h of treatment. The percentage of late-stage apoptotic cells increased, after treatment with the extract at higher concentrations (0.125 and 0.25 mg/mL). A good effect was already observed at 0.125 mg/mL, where, after treatment, 6.42% of cells were in early-apoptotic stage and 9.09% were in the late-apoptotic stage. At a concentration of 0.25 mg/mL, Fraction D induced a strong cytotoxic effect determining the initiation of the necrotic process (Figure 3F,G).

These findings suggest that Fraction D is responsible for TR anticancer action in K562 cells.

### 3.4. Fraction D Inhibits BCR-ABL/STAT5

To confirm the key role of Fraction D in inhibiting BCR/Abl, K562 cells were treated with all four fractions as well as with the total polyphenol extract at 0.1 mg/mL for 24 h. Only Fraction D, which was found to be the most toxic of all fractions, reduced the anti-pY247AB expression levels (Figure 4A). However, no modulation of BCR/Abl at mRNA level was observed until 48 h of treatment (Appendix A). To validate this finding, BCR-ABL/STAT5 signaling pathway modulation was analyzed in K562 cells treated for 24 and 48 h with 0.1 mg/mL of Fraction D (Figure 4B). Fraction D totally abolished constitutive phosphorylation of BCR in Y177 as well as in Y247 at 0.1 mg/mL, for 48 h, with a significant reduction already at 24 h (Figure 4B,C). Although a slight increase in phosphorylation status of its downstream targets was observed at 24 h, the total inactivation of BCR/Abl at 48 h was associated with abrogation of anti-pY694STAT5 and its total levels, as well as reduced levels of both anti-pY207CRKL and AKT. Notably, a significant decrease in these targets was observed by comparing their levels at 24 and 48 h. We hypothesized that Fraction D is able to target BCR/Abl, blocking its phosphorylation at an early stage, leading to a block of downstream signals at a later stage. This mechanism of action is in full agreement with the strong antiproliferative effect of Fraction D at 0.125 mg/mL, at 48 h, which induces 60% cell death (Figure 3D,E). In contrast, treatment with Fraction D promoted p38 expression >16-fold after 24 h and >7-fold after 48 h, compared to the control.

Taken together, these findings indicate that Fraction D is responsible for BCR-ABL/STAT5 inhibition and p38 signaling pathway activation in CML cells.

### 3.5. Identification of Molecules in Fraction D

Fraction D was characterized by RP–UHPLC–MS/MS (Figure 5A) and nine compounds were identified (Figure 5B,C), including coumarins, isoflavone glucosides, and aglycones. Polyphenol identification was carried out on the basis of retention times and by comparing mass spectra and MS/MS data with those reported in the literature.

Among the isoflavones identified, the most abundant aglycone was found to be Medicarpin (**#**7; Figure 5A,B). This pterocarpan showed two intense fragment ions at m/z 161 [M-H-C_6_H_6_O_2_]^+^ and at m/z 254, due to the loss of a methyl group [M-H-CH_3_]^−^. The second most intense peak (#4;**Figure 5**A,B), was identified as Daphnoretin, which showed two fragment ions at m/z 336 and m/z 191, due to the loss of methyl and C_4_-O-C_7_ groups [M-160-H]^−^, respectively. Peak 1 (Figure 5A,B) was identified as Glucosyl or galactosyl-malonyl-Formononetin, which showed an intense fragment ion at m/z 269, corresponding to the loss of the sugar moiety. Peak 5 (Figure 5A,B) showed two main fragment ions at m/z 213 [M-H-CH_3_]^+^ and m/z 254 [M-H-CO-CO]^+^, and was tentatively identified as Formononetin.

### 3.6. Effect of Isoflavonoid Trifolium Repens Fraction on Leukemia Cells

To assess the selectivity of this isoflavonoid portion of TR in CML, compared to other leukemias, we evaluated its effects in U937 and NB4 cells treated for 24 and 48 h, at different concentrations ranging from 2.0 mg/mL to 0.035 mg/mL (Figure 6A,B). As expected from an isoflavonoid-rich extract, Fraction D induced a block in both U937 and NB4 cell proliferation not seen in the total extract (Figure 1A,B), which also contains non-active molecules. For this reason, at the dose of 0.25 mg/mL, the total extract did not affect cell viability, while at the same dose Fraction D induced robust cell death at about 50% and 70% in U937 and NB4 cells, respectively. However, the antiproliferative effect of Fraction D in K562 cells was stronger than that observed in these two cell lines. IC_50_ values at 48 h were 0.23 mg/mL (2.5-fold higher than K562) for U937 and 0.124 mg/mL (1.35-fold higher than K562) for NB4 cells (Figure 6A,B), versus 0.092 mg/mL for K562 cells (Figure 3D). This different cellular behavior might be attributed to the key role of BCR/Abl in amplifying downstream targets. In line with this hypothesis, Fraction D at 0.1 mg/mL induced a stronger reduction in *STAT5* gene expression in K562 cells than in U937 and NB4 cells, after 24 and 48 h of treatment (Figure 6C), underscoring its greater selectivity in CML.

Notably, Fraction D induced a weak antiproliferative effect in Mepr2B cells, [44], an immortalized normal cell line (Figure 6D). Only the highest dose (2.0 mg/mL) induced a block in normal proliferation, suggesting that Fraction D exerts a cancer-selective antiproliferative effect.

To better investigate the specificity of the effect of *T. repens* extract on the BCR-ABL/STAT5 pathway, normal Mepr2B cells were treated with Fraction D at 0.1 mg/mL for 24 and 48 h (Figure 6E,F). Treatment at 0.1 mg/mL of these cells, which do not carry the BCR/Abl hybrid protein (Figure 6E,F) did not significantly affect the levels of pY308AKT, AKT, pY694STAT5, and pThr180/Tyr182p38, at 24 and 48 h, compared to the K562 cells.

Taken together, these findings indicate that the antiproliferative effects of Fraction D were triggered by the fusion protein BCR/Abl, leading to inhibition of the STAT5 pathway.

To identify the active compound responsible for the antiproliferative effect of Fraction D, the three most abundant peaks were isolated and tested in K562 cells. Based on their relative percentage in Fraction D, the individual molecules Formononetin (peak #5; 7% of Fraction D), Daphnoretin (peak #4; 14%), and Medicarpin (peak #7; 26%) were tested at the corresponding concentration of Fraction D at 0.5 mg/mL for 48 h (Figure 7A,B). No toxicity was observed after Daphnoretin and Formononetin treatment, which induced only 20% of cell death, while Medicarpin was able to induce a robust cell death at a similar level (80%) to the total extract, as shown by PI analysis and cell imaging (Figure 7A,B).

Our findings suggest that Medicarpin might be the molecule contained in the natural extract responsible for antitumor activity. However, further investigations will be required to better characterize and optimize its properties.

## 4. Discussion

CML is caused by a genetic alteration of the Ph chromosome, which forms the hybrid protein BCR/Abl [5,6]. BCR/Abl is responsible for JAK/STAT, PI3K/AKT, and CRKL expression, and consequently CML progression [7,8,9]. Inhibiting BCR-ABL expression has long been the gold-standard approach in CML treatment.

Polyphenols are abundantly present in clover and constitute an important group of phytochemical compounds extensively used in medicine [26]. Several species of clover have displayed beneficial effect in human diseases. TR, commonly known as white clover is used as a fodder crop for cattle, but to date, its potential anticancer activity has not been explored.

In this study, we investigated the antitumor action of TR in several cancer cell lines, focusing specifically on its effect in CML cells. We found that TR total extract was able to induce cancer cell death in a dose-dependent manner in K562 cells, but not in other cancer cell lines. These results prompted us to evaluate the effects of TR total extract on BCR-ABL/STAT5 pathway, known to be responsible for CML progression. We observed a strong reduction in BCR/Abl phosphorylation levels, after 24 and 48 h of TR treatment at 0.5 mg/mL, suggesting that the polyphenol extract functions via a specific mechanism. TR induced a decrease in phosphorylation of components of the indicated pathway already at 24 h, during which time signal transduction was initiated on these specific targets. However, the later occurring cell death at 48 h might be explained by the fact that the final readout of proliferation inhibition is a phenotypic finding, and at this concentration, it is likely that signal transduction is starting to drive cell death at a later point. This result is also supported by the concentration of 1.0 mg/mL, which is able to induce cell death ≥40%. In agreement, concentrations lower than 0.5 mg/mL indicate that there is yet no regulation of phosphorylation of these targets, suggesting that the effects start at a dose of 0.5 mg/mL, around 24 h (Appendix A).

By separating and analyzing four different fractions, we demonstrated that the active portion of TR on CML cells belongs to a class of isoflavonoids that is already known for its beneficial properties [29,30,31].

Some of the molecules present in this fraction are reported to regulate the expression of anti-apoptotic proteins, such as JAK/STAT and PI3K/AKT, expressed in numerous leukemias. Formononetin induces apoptosis in HeLa cancer cell lines, inhibiting AKT phosphorylation in vitro and in a xenograft model [45]. Using xenograft models of breast cancer, Formononetin showed growth-inhibitory activity associated with the inhibition of tumor angiogenesis [46] and mTOR pathway [47]. This compound also inhibits multiple myeloma cell proliferation and induces apoptosis by suppressing STAT3 and STAT5 activation [48]. Calycosin (#2; Figure 5A,B) acts on the PI3K/AKT/mTOR pathway, blocking tumor growth and inducing apoptosis in an estrogen receptor-positive osteosarcoma cell line [49]. Daphnoretin inhibits E6 protein expression [50] and arrests cell cycle in the G2/M phase, by down-regulating Cdc2, cyclin A, and cyclin B1 [51]. It also blocks Bcl-2 expression and induces Bax expression [51]. Medicarpin inhibits NF-κB signaling by attenuating TNFα-induced nuclear translocation of p65 [52].

In this study, we found for the first time that TR exerts antitumor effects in CML. Reduced survival of K562 cells after exposure to total extract and Fraction D is associated with inhibition and activation of BCR-ABL/STAT5 and p38 signaling pathways, respectively. These biological effects are mediated by its isoflavonoid-rich portion and are greater in cells harboring the fusion protein (K562) than in cells that do not carry this translocation (U937 and NB4). Isolation of the isoflavonoid-rich fraction increases the concentration of active molecules, compared to the TR total extract (Figure 1F,G), thus, corroborating and strengthening that Fraction D (and the molecules contained therein) is responsible for the enhanced biological effect that can explain this result (Figure 6A,B).

The isoflavonoid fraction of TR, Fraction D, was able to drastically reduce the levels of BCR/Abl protein expression and, consequently, selectively inhibit all downstream pathways, with an IC_50_ of 0.092 mg/mL in K562 cells. Excitingly, this fraction displayed low toxicity in normal cells, potentially making it an excellent option for chemotherapy.

Lastly, we identified Medicarpin as the molecule responsible for the antiproliferative action of Fraction D. Specifically, this drug was able to induce strong CML cell death to the same extent as Fraction D (Figure 7A,B). Although further investigations will be required to obtain a better biological characterization of Fraction D/Medicarpin and to test their antitumor efficacy and toxicity in vivo, our preliminary findings suggest that the development and biochemical optimization of phytochemical molecules contained in *T. repens* might lead to the identification of therapeutic agents active against CML.

## Figures and Tables

**Figure 1 cells-09-00379-f001:**
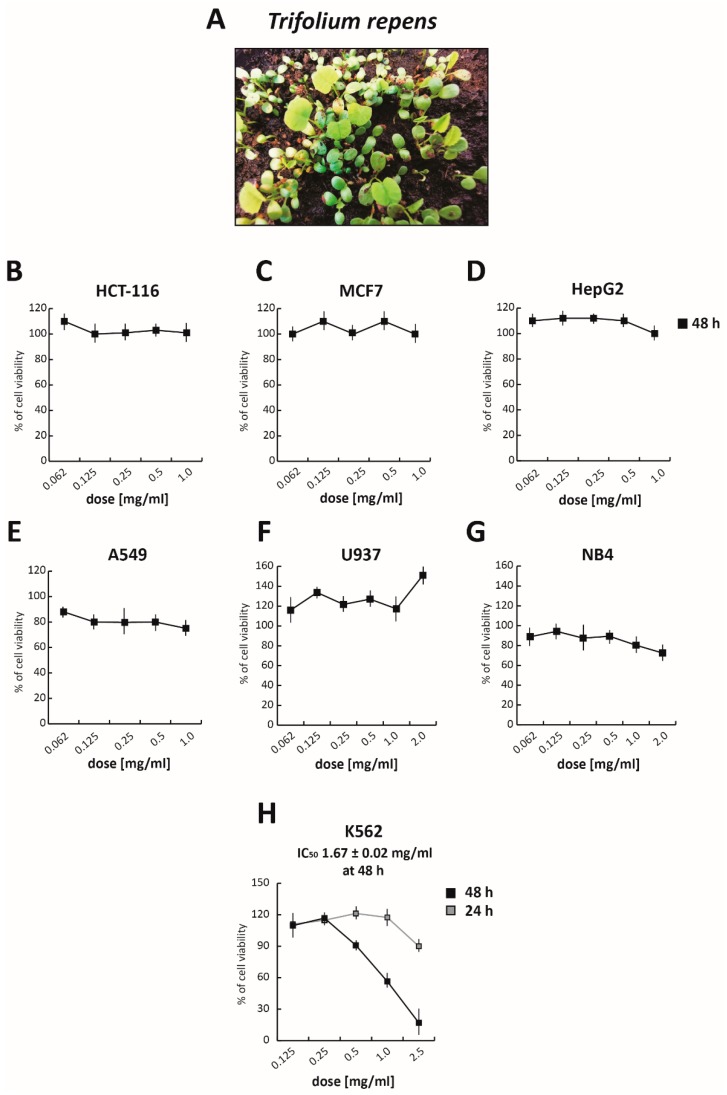
*Trifolium repens* (TR) inhibits cancer proliferation. (**A**) Image of TR meadow at 5 days. (**B**–**H**) Cell growth rates determined by MTT assay of (**B**) HCT-116, (**C**) MCF7, (**D**) HepG2, (**E**) A549, (**F**) U937, (**G**) NB4 cells after treatment with TR for 48 h at the indicated concentrations, and (**H**) K562 cells treated with TR at different doses at 24 and 48 h; IC_50_ at 48 h, was determined. Values are mean ± SD of biological triplicates.

**Figure 2 cells-09-00379-f002:**
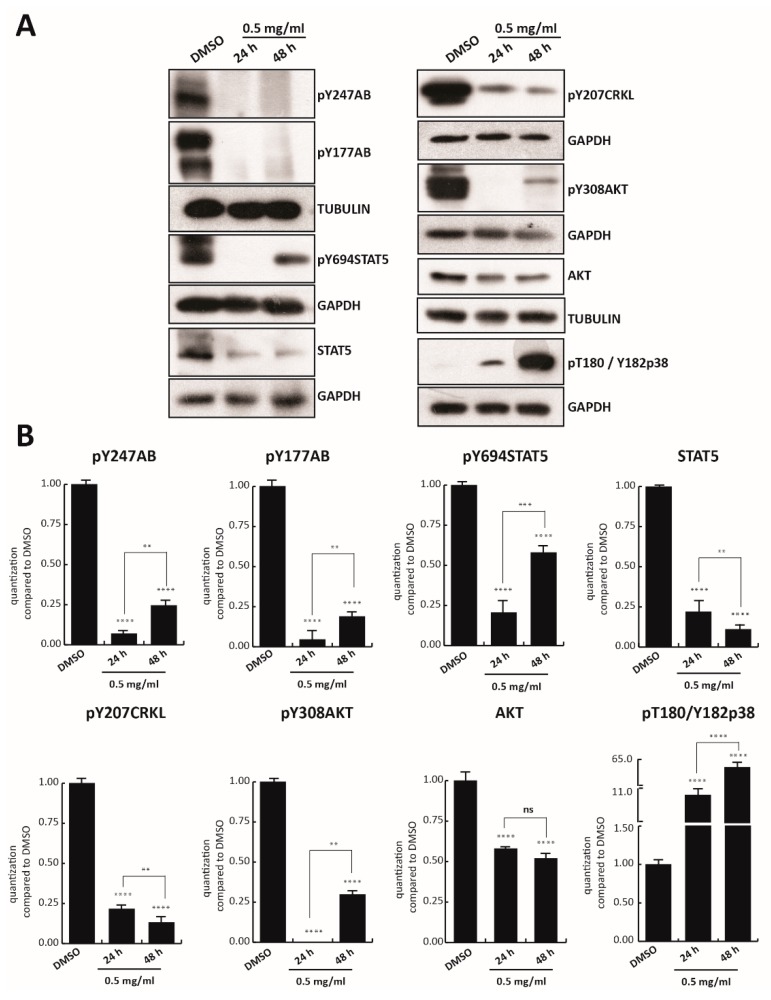
TR inhibits BCR-ABL/STAT5 signaling pathway. (**A**) Western blot analysis of pY247AB, pY177AB, pY694STAT5, STAT5, pY207CRKL, pY308AKT, AKT, and pT180/Y182p38. K562 cells were treated with TR total extract for 24 and 48 h at 0.5 mg/mL; (**B**) Normalization of protein level expression using the ImageJ software. Values are mean ± SD of biological duplicates. **** *p*-value ≤ 0.0001, *** *p*-value ≤ 0.001, ** *p*-value ≤ 0.01, * *p*-value ≤ 0.05, ns *p*-value > 0.05 vs. control cells.

**Figure 3 cells-09-00379-f003:**
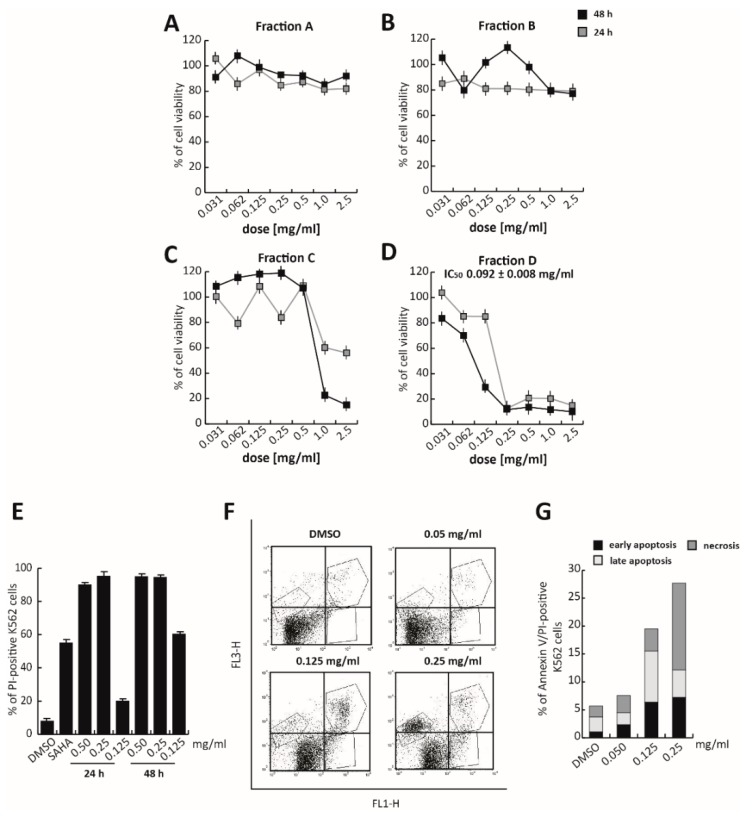
Antiproliferative effects of Fraction D in K562 cells. K562 cells were treated for 24 and 48 h with (**A**) Fraction A, (**B**) Fraction B, (**C**) Fraction C, and (**D**) Fraction D, at seven different concentrations from 2.0 mg/mL to 0.031 mg/mL. (**E**) K562 cell death analysis by propidium iodide (PI), after treatment with Fraction D at the indicated times and concentrations. SAHA (N’-hydroxy-N-phenyloctanediamide; Vorinostat) was used at 5 μM, for 24 h, as a positive control. (**F**, **G**) K562 apoptotic cell death analysis by Annexin V/PI experiments after treatment with Fraction D at the indicated concentrations for 24 h. Values are mean ± SD of biological triplicates.

**Figure 4 cells-09-00379-f004:**
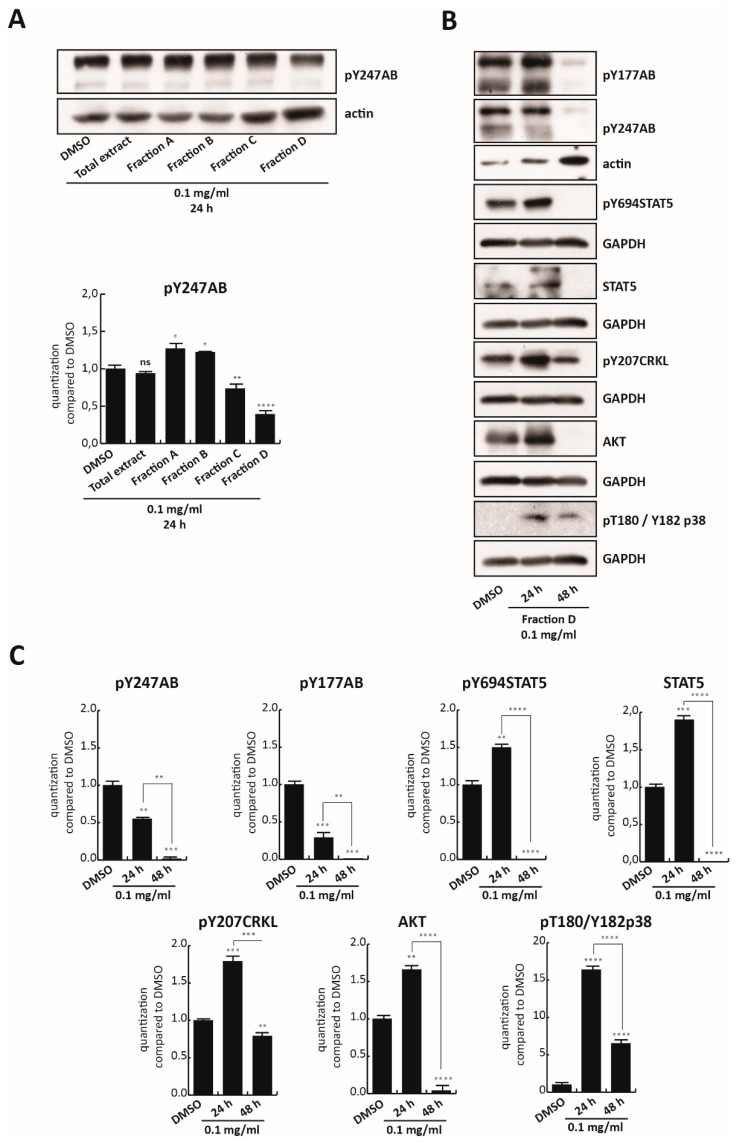
Fraction D inhibits BCR-ABL/STAT5. (**A**, upper panel) Western blot analysis of pY247AB expression levels in K562 cells after 24 h treatment with TR total extract and all four fractions at 0.1 mg/mL. (**A**, lower panel) Graphic normalization by ImageJ software. (**B**) pY247AB, pY177AB, pY694STAT5, STAT5, pY207CRKL, AKT, and pT180/Y182p38 expression levels in K562 cells treated with Fraction D at 0.1 mg/mL for 24 and 48 h. (**C**) Normalization of protein level expression by the ImageJ software. Values are mean ± SD of biological duplicates. **** *p*-value ≤ 0.0001, *** *p*-value ≤ 0.001, ** *p*-value ≤ 0.01, * *p*-value ≤ 0.05, ns *p*-value > 0.05 vs. control cells.

**Figure 5 cells-09-00379-f005:**
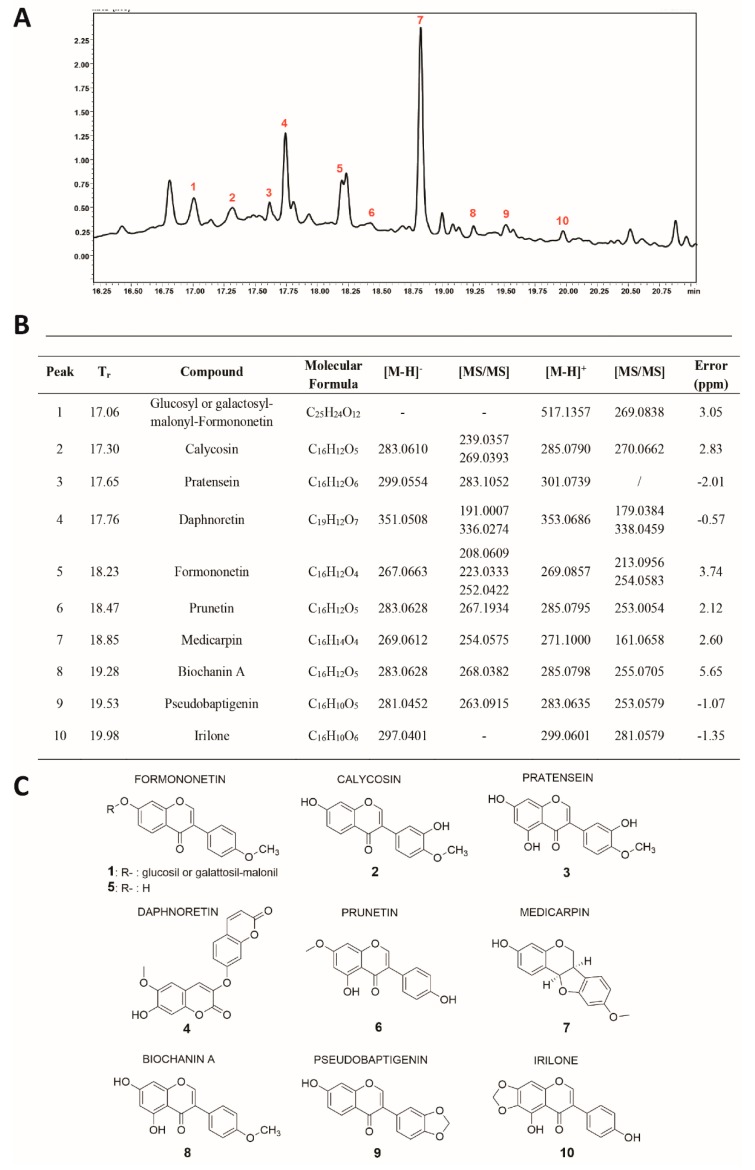
(**A**) Chromatographic profile (λ: 280 nm), (**B**) list, and (**C**) chemical structures of all polyphenol compounds detected in Fraction D of the TR extract.

**Figure 6 cells-09-00379-f006:**
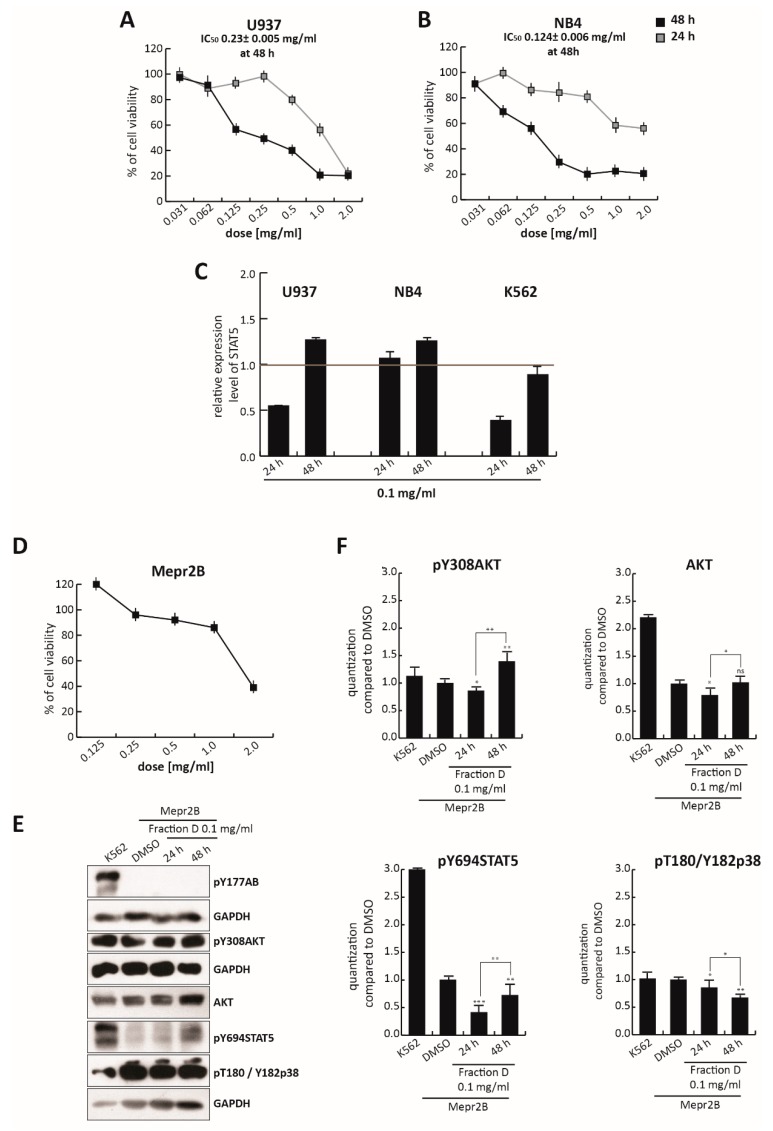
Effect of Fraction D in leukemia and normal cell lines. Cell growth rates determined by MTT of (**A**) U937 and (**B**) NB4 cells treated with Fraction D for 24 and 48 h, at the indicated concentrations, and of (**D**) Mepr2B cells challenged with Fraction D for 48 h, at the indicated concentrations. (**C**) *STAT5* gene expression analysis of U937, NB4, and K562 cells treated with Fraction D at 0.1 mg/mL, for 24 and 48 h. Values are mean ± SD of biological triplicates. (**E**) pY177AB, pY694STAT5, pY308AKT, AKT, and pT180/Y182p38 expression levels in Mepr2B cells treated with Fraction D at 0.1 mg/mL for 24 and 48 h. (**F**) Normalization of protein level expression by the ImageJ software. Values are mean ± SD of biological duplicates. **** *p*-value ≤ 0.0001, *** *p*-value ≤ 0.001, ** *p*-value ≤ 0.01, * *p*-value ≤ 0.05, ns *p*-value > 0.05 vs. control cells.

**Figure 7 cells-09-00379-f007:**
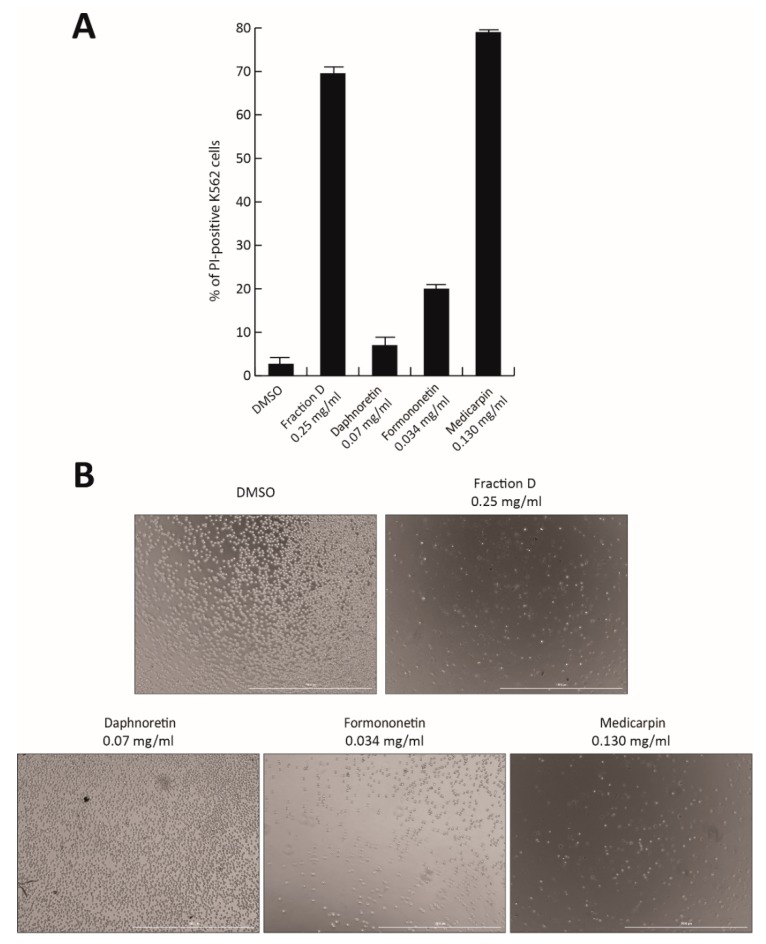
Antiproliferative effect of separate components of Fraction D. (**A**) K562 cell death analysis by propidium iodide (PI) after treatment with Fraction D, Daphnoretin, Formononetin, and Medicarpin for 48 h, at the indicated concentrations. (**B**) Images of K562 cells after treatment with Fraction D, Daphnoretin, Formononetin, and Medicarpin for 48 h, at the indicated concentrations.

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
