# Peer review of "Trifolium Repens Blocks Proliferation in Chronic Myelogenous Leukemia via the BCR-ABL/STAT5 Pathway"

_cells, 2020, doi:10.3390/cells9020379_

Round 1

Reviewer 1 Report

In this manuscript,  Federica et al analyzed the anticancer effects of the extracts from T. repens in cancer cell lines. They found that the total extract and its isoflavonoid-rich fraction showed cytotoxicity in CML K562 cells and they also found the possible mechanism underlying the cell proliferation arrest is inhibition of BCR-ABL/STAT5 and activation of p38 pathways. This is an interesting study, the following suggestions may strengthen the manuscript.

Major points

In the fraction D, it contains 10 compounds and 9 is identified, one seems an unknown compound, what is the chemical structure of the unknown compound?  is it a new compound? Also, it is better to test the effect on cell proliferation arrest for every single compound and identify who is responsible for the effect? Or only the extract has the effect. Many extracts from natural products have a better anti-tumor effect in vivo, which not only through direct cytotoxicity but also through regulation of immunological function. It is better to have the in vivo data for the extracts or single compounds. Does the extract inhibit BCR-ABL/Stat specifically? It is better to provide the data in Mepr2B cells.

Minor point

  It has many targeted drugs for BCR-ABL. It is better to discuss more on what is the benefit of developing the new compound from natural products.

Author Response

In the fraction D, it contains 10 compounds and 9 is identified, one seems an unknown compound, what is the chemical structure of the unknown compound?  is it a new compound?

We thank the Reviewer for this observation. Fraction D contains 10 compounds, as reported in the chromatogram (Figure 5A) and the table (Figure 5B), all of which were identified. Figure 5C originally reported only 9 chemical structures since the structure of Formononetin is repeated twice (peak #1 and #5): in peak #1 the molecule is linked to a sugar and in peak #5 it is free. We have modified Figure 5C, which now reports the corresponding numbers below the chemical structures.

Also, it is better to test the effect on cell proliferation arrest for every single compound and identify who is responsible for the effect? Or only the extract has the effect.

Following the Reviewer’s comments, we isolated and tested the three principal peaks (#4, #5, #7) in Fraction D (Daphnoretin, Formononetin, and Medicarpin, respectively). These components were tested on K562 cells for 48 h in an attempt to identify the molecule responsible for antitumor activity. The doses used for treatment were chosen based on the relative percentages that the molecules present in Fraction D at 0.5 mg/ml (Figure 7A, B). No toxic effect was observed after treatment with Daphnoretin and Formononetin, while Medicarpin, the most abundant peak, induced significant cell death, comparable to the total extract. We have modified the text accordingly in the Results section (lines 437-447) and in the new Figure 7.

Many extracts from natural products have a better anti-tumor effect in vivo, which not only through direct cytotoxicity but also through regulation of immunological function. It is better to have the in vivo data for the extracts or single compounds.

We agree with the Reviewer that the anticancer effect of natural compounds often also results from the regulation of immunological functions. In fact, natural products are known to have numerous beneficial therapeutic effects since they are able to act simultaneously on specific altered targets and on the immune system. In vivo studies will indeed be needed to better evaluate the activity of these molecules, but fall outside the scope of this work. Here, we report preliminary data that describe the principal molecular target of T. repens extract. Before performing in vivo investigations, pharmacokinetic and pharmacodynamic studies will also be necessary to better understand its biological effect.

Does the extract inhibit BCR-ABL/Stat specifically? It is better to provide the data in Mepr2B cells.

We thank the Reviewer for their comment. Based on the Reviewer’s suggestion, we investigated whether the Trifolium extract preferentially inhibits the BCR-ABL/SATAT5 pathway by evaluating its effects on Mepr2B cells, which do not carry the fusion protein BCR/Abl. Our results, now described in subsection 3.6. Effect of Isoflavonoid Trifolium Repens Fraction on Leukemia Cells, show that Fraction D specifically inhibits BCR-ABL/STAT5 signaling pathway, not reducing the level of proteins involved in cell proliferation such as phAKT, AKT, phSTAT5 and php38 in Mepr2B (lines: 416-422; Figure 6E, F).

Minor point

  It has many targeted drugs for BCR-ABL. It is better to discuss more on what is the benefit of developing the new compound from natural products

This is a good point. We have now discussed this issue in the Introduction, focusing on the advantages of using molecules from natural extracts (lines: 61-78).

Reviewer 2 Report

This is a well-designed and well-carried out study. That the effects of cell treatment with Trifolium repens are CML-restricted is of high interest. The Supplementary Material is extremely difficult to follow.

The reason why the Authors showed dose/response curves in two separate Figures (1 and 2) is not clear. The differences appear to be limited to the highest concentration (1 mg/ml in Figure 1 and 2/2.5 mg/ml in Figure 2) or to the presence of an additional time of incubation (24 h) for K562 cells in Figure 2C. On the other hand, if the idea was to group all the haematopoietic cells together, Figure 1F should be moved to Figure 2, but in this case Figure 1F would be redundant with Figure 2C. This Reviewer suggests to show the picture of TR alone (maybe enlarged a bit) as Figure 1 and group all the graphs together as a new Figure 2 (eliminating Figure 1F or merging the data reported there with those of Figure 2C).

Line 175 - it should be “…positive control for the induction of cell death in our experimental system.”

Line 277 - “…we first investigated the effect of TR on BCR-ABL expression.” The expression of BCR/Abl fusion protein is usually determined using anti-cAbl AB. On the contrary, the experiment shown refers to BCR/Abl phosphorylation, as determined by using anti-pY177 AB. Similar sentences, like that at line 341, “reduced pY247-BCR-ABL expression levels” should be avoided. The text should be corrected accordingly throughout.

Line 282 - Figure S1 apparently reports TRdose/response relationships for BCR/Abl and Crkl, but not Akt, phosphorylation.

Figure S1 - There is a good progression of effect going from DMSO-treated control to TR 0,25 and to 0,125 mg/ml. The treatment with TR 0,5 mg/ml shows a discrepancy between the effect on BCR/Abl phosphorylation and that on Crkl phosphorylation. The Authors should make a comment about such a discrepancy, which is, however, relatively often observed and therefore does not hamper the interpretation of the overall outcome of the experiments. This is also in keeping with the results shown in Figure 3A, where TR treatment suppresses completely BCR/Abl phosphorylation in Y177/247 and strongly reduces, but does not suppress, Crkl phosphorylation in Y207.

Lines 325-328 - Why is the emphasis on the apoptotic cell fractions while the overall cell death is higher at 0,25 that 0,125 mg/ml TR? What is wrong with the induction of necrosis instead of apoptosis? In this Reviewer’s opinion, what it is important is the capacity of treatment to kill cells.

Line 325 - it should be “…mg/ml, where, after treatment, 6.42…” (add comma).

Figure 7 - How do the Authors reconcile the idea that the effects of cell treatment with Trifolium repens are CML-restricted with the results relative to U937 and NB4 cells? The point is in part addressed in lines 444-445, but the Authors should perhaps make stronger statements.

Author Response

The reason why the Authors showed dose/response curves in two separate Figures (1 and 2) is not clear. The differences appear to be limited to the highest concentration (1 mg/ml in Figure 1 and 2/2.5 mg/ml in Figure 2) or to the presence of an additional time of incubation (24 h) for K562 cells in Figure 2C. On the other hand, if the idea was to group all the haematopoietic cells together, Figure 1F should be moved to Figure 2, but in this case Figure 1F would be redundant with Figure 2C. This Reviewer suggests to show the picture of TR alone (maybe enlarged a bit) as Figure 1 and group all the graphs together as a new Figure 2 (eliminating Figure 1F or merging the data reported there with those of Figure 2C).

We agree with the Reviewer. We have now modified the figures, grouping Figures 1 and 2 in the new Figure 1, except for the graph relating to K562 (original Figure 1F), which is now reported in Figure S1A.

Line 175 - it should be “…positive control for the induction of cell death in our experimental system.”

We have corrected the text as suggested (now line 186-187).

Line 277 - “…we first investigated the effect of TR on BCR-ABL expression.” The expression of BCR/Abl fusion protein is usually determined using anti-cAbl AB. On the contrary, the experiment shown refers to BCR/Abl phosphorylation, as determined by using anti-pY177 AB. Similar sentences, like that at line 341, “reduced pY247-BCR-ABL expression levels” should be avoided. The text should be corrected accordingly throughout.

We thank the Reviewer for highlighting this point. We have modified the text throughout accordingly.

Line 282 - Figure S1 apparently reports TRdose/response relationships for BCR/Abl and Crkl, but not Akt, phosphorylation.

Again, we thank the reviewer for bringing this to our attention. We have now expanded Figure S1B by adding the levels of pY308AKT after TR treatment (Figure S1B).

Figure S1 - There is a good progression of effect going from DMSO-treated control to TR 0,25 and to 0,125 mg/ml. The treatment with TR 0,5 mg/ml shows a discrepancy between the effect on BCR/Abl phosphorylation and that on Crkl phosphorylation. The Authors should make a comment about such a discrepancy, which is, however, relatively often observed and therefore does not hamper the interpretation of the overall outcome of the experiments. This is also in keeping with the results shown in Figure 3A, where TR treatment suppresses completely BCR/Abl phosphorylation in Y177/247 and strongly reduces, but does not suppress, Crkl phosphorylation in Y207.

We agree with the Reviewer that a discrepancy between the effect on BCR/Abl phosphorylation and on Crkl phosphorylation is often observed. In our case, however, this discrepancy was not found. Specifically, any apparent discrepancy is due to different basal levels of these two targets: constitutive phosphorylation of CRKL is higher that of BCR/Abl. Indeed, when we quantified the band intensity, protein levels were comparable (see graphs as reported in word file attached referring to Figure S1B). Similarly, we do not see any significant differences in protein levels relating to Figure 2

Lines 325-328 - Why is the emphasis on the apoptotic cell fractions while the overall cell death is higher at 0,25 that 0,125 mg/ml TR? What is wrong with the induction of necrosis instead of apoptosis? In this Reviewer’s opinion, what it is important is the capacity of treatment to kill cells.

We agree with the Reviewer’s comment. We have now changed the sentence in the text (line: 341).

Line 325 - it should be “…mg/ml, where, after treatment, 6.42…” (add comma).

We have modified the text as suggested (line 341).

Figure 7 - How do the Authors reconcile the idea that the effects of cell treatment with Trifolium repens are CML-restricted with the results relative to U937 and NB4 cells? The point is in part addressed in lines 444-445, but the Authors should perhaps make stronger statements.

Following the Reviewer’s suggestion, we have now addressed this point further both in the Results (lines 407-410) and Discussion section (lines: 496-498).

Round 2

Reviewer 1 Report

My concerns are well answered in the revised manuscript. Hopefully anti-tumor effect will be tested in vivo in future.